# Towards Sustainable Dairy Production in Argentina: Evaluating Nutrient and CO₂ Release from Raw and Processed Farm Waste

**Gastón A. Iocoli** [1,2], **Luciano Orden** [1,3], **Fernando M. López** [1,2], **Marisa A. Gómez** [1], **María B. Villamil** [4,*] and **María C. Zabaloy** [1,2,*]

1   Departamento de Agronomía, Universidad Nacional del Sur (UNS), San Andrés 800,
    Bahía Blanca 8000, Argentina; gastoniocoli@yahoo.com.ar (G.A.I.); luciano.orden@uns.edu.ar (L.O.);
    fernando.lopez@uns.edu.ar (F.M.L.); manahigomez@gmail.com (M.A.G.)
2   Consejo Nacional de Investigaciones Científicas y Técnicas (CONICET) Centro de Recursos Naturales
    Renovables de la Zona Semiárida (CERZOS), Departamento de Agronomía, Universidad Nacional del Sur,
    San Andrés 800, Bahía Blanca 8000, Argentina
3   Instituto Nacional de Tecnología Agropecuaria (INTA), Estación Experimental Agropecuaria Hilario
    Ascasubi, Ruta 3 km 794, Ascasubi 8142, Argentina
4   Department of Crop Sciences, University of Illinois at Urbana Champaign, 1102 S. Goodwin Ave.,
    Urbana, IL 61801, USA
*   Correspondence: villamil@illinois.edu (M.B.V.); mzabaloy@uns.edu.ar (M.C.Z.)

**Abstract:** Mineralization studies are the first step in determining the usefulness of an amendment such as fertilizer, and are essential to creating guidelines for dairy waste management to help producers make informed decisions. Our goal was to assess the effects of dairy raw, composted, and digested manure amendments on C, N, and P mineralization to evaluate the feasibility of their in-farm production and use as organic fertilizers. The liquid and solid fractions of dairy effluent (LDE, SDE), dairy effluent digestate (DED), onion–cattle manure digestate and compost (OCMD, OCMC) were characterized by chemical and spectroscopic methods. Soil microcosms with LDE, SDE, DED, OCMD and OCMC and the C, N and P mineralization were determined periodically. Elemental and structural differences among amendments led to contrasting profiles of C, N, and P mineralization, and thus to differences in nutrient availability, immobilization, and CO₂ emission. All processed materials were more stable than untreated waste, reducing C emissions. Digestates showed net C immobilization, and supplied the highest levels of available N, creating a relative P deficit. Instead, the compost supplied N and P via mineralization, producing a relative P excess. Future studies should aim at evaluating fertilization strategies that combine both kinds of amendments, to exploit their complimentary agronomic characteristics.

**Keywords:** carbon mineralization; nitrogen mineralization; phosphorous mineralization; dairy manure; compost; onion waste; dairy effluents

## 1. Introduction

The management of waste in dairy farming is associated with inefficiencies in the cycling of nutrients, leading to the environmental pollution of watercourses. The situation is pressing in developing countries such as Argentina, where the lack of legislation and guidelines, along with the low adoption rates of waste management technologies, accompany the accelerated consolidation of dairy farms triggered by slim margins of production and 40% annual inflation rates [1,2]. These are critical issues the Instituto Nacional de Tecnología Agropecuaria (INTA), the governmental agency responsible for agro-technology transfer, is addressing with dairy farmers throughout the country.

Argentina has a long tradition of dairy farming, ranking 18th in the world and third within South America in the production of fluid milk, contributing about 1.6% of the global production [2,3]. In 2020, the official statistics reported a total milk production of 11.35 million metric tons coming from 10,411 dairy operations and 1.6 million dairy cows.

Most dairy farms are small farms, predominantly pasture-based with year-round cattle grazing supplemented with compound feed, and an average stocking-rate of 1.4 cows per hectare, in non-irrigated land [4]. Dairying is seldom exclusive to an operation, and beef and grain farming commonly accompanies a rotational forage dairy system. Following the global trend in this industry, the current slim economic margins and growing inflation rates, however, dairy farms in Argentina are undergoing a process of consolidation and intensification, evidenced by a reduction on the number of farms and a parallel increase in cattle productivity and herd size over the last 20 years [2,4]. At the same time, non-grazing systems have gained traction, currently accounting for 16% of the dairy operations. These systems are mainly dry-lots, where cows are kept in open lots, with some free stalls and compost barns as well. In the main area of production, the Pampas region, Lazzarini et al. [4] reported a typical dairy cow diet including 45% pasture, 26% silage, and 29% concentrate feeds (maize grain and soybean meal). Feeding of silage and concentrates occurs in part-time confinement while cows are milked; milking, in turn, occurs two to three times per day. With the parlor cleaned after each session, the average effluent volume used amounts to 27.7 L $day^{-1}cow^{-1}$, estimated to add up to 30,000 million L $year^{-1}$ [1]. Effluents from milking and cleaning are sent to nearby ponds; stored effluents are used for agronomic purposes in 60% of the farms without any further processing [1,4].

Due to the aforementioned changes in the dairy industry, however, total waste and wastewater discharge are expected to increase significantly, requiring adequate on- and off-farm management. Although there is no current national legislation on effluent management, specific regulations of limited geographical scope have started to appear, driven by pollution events and societal concerns [4]. According to a recent survey on the dairy manure management perceptions and needs of South American farmers by Herrero et al. [1], respondents were generally well aware of the potential use of manure as fertilizer and for biogas production, and of the consequences of the mismanagement of effluents for water contamination and pathogen transmission. Yet when surveyed about their needs and barriers to the adoption of waste management strategies and technologies, farmers requested guidelines for responsible management, indicating that a lack of knowledge was one of the top barriers to technology adoption. Across Argentina, INTA researchers work closely with farmers to generate these guidelines with a scope that fits producers' and regional needs. As a result, INTA H. Ascasubi, in the southern semiarid Pampas region, collaborated with a local dairy farmer and university personnel to study different manure and slurry treatments in order to keep the nutrients in the farm and out of the water bodies.

Manure and other agro-industrial wastes used as soil amendments without further processing are extremely unstable materials with potential for leaching and/or volatilization [5]. Although composting is an acceptable option to reduce their volume and stabilize these materials, anaerobic digestion is currently considered a better alternative, as it combines residue treatment with the production of renewable energy [6]. The degradation of organic wastes with anaerobic digestion reduces methane gas emission, pathogens loads, and unpleasant odors released to the environment, while producing a nutrient-rich fertilizer [7]. The sustainability of the production of biogas itself depends on the proper use of the digested material, which must be treated, disposed of, or reused properly, so as to avoid any negative environmental impact [8]. The use of the digested material as an organic fertilizer in agriculture seems to be an optimal solution, since it contains significant amounts of organic matter and nutrients available for plants [9,10]. The mineralization of these amendments releases large amounts of $CO_2$ (due to the activation of the soil microbiota), nitrogen (N), phosphorus (P), and other nutrients, yet the availability and timing of these nutrient releases depends on the biodegradability of the organic matter that characterizes the amendment [11–13]. Thus, mineralization studies are the first step towards determining the usefulness of an amendment, such as fertilizer [14].

The balance between the immobilized and mineralized N following the addition of organic amendments to the soil regulates N availability for plants [15]. In turn, this balance is determined by the C/N ratio of the soil amendments, the structure of the organic

compounds that bound the N, and the interaction with soil properties [6]. Concerning P, the availability of organic P may be greater than that of C and N [16]. In most cases, microorganisms use organic P compounds mainly as a source of C and will only incorporate a small portion of the P released [17], thus increasing P availability during the degradation of amendments. Other processes also affect the P availability in soil, such as adsorption/desorption and precipitation/dissolution [18], related to textural class, soil pH, and organic matter content associated with the mineral fraction [19].

This study, resulting from the collaboration among INTA, university personnel, and a local dairy farm, evaluates the mineralization potential of dairy waste in unprocessed material (solid fraction of dairy effluent, SDE; liquid fraction of dairy effluent, LDE) and processed forms (dairy effluent digestate, DED; cattle manure co-digested with onion residues from a local processing plant, OCMD; cattle manure co-composted with onion residues, OCMC) over a period of time equivalent to the growing season of onion (*Allium cepa* L.), an important crop of the region. Thus, our goal was to characterize the release of $CO_2$, N, and P from raw, composted, and digested manure amendments to evaluate the feasibility of their use as organic fertilizers in the agricultural systems of southwest Buenos Aires. The availability of this information will contribute to creating the necessary guidelines for dairy waste management and help producers in the region to make informed decisions.

## 2. Materials and Methods

### 2.1. Dairy Farm Description

The study was conducted at a commercial dairy farm located near Hilario Ascasubi (39°17′S, 62°32′ W), Buenos Aires Province, Argentina. The climate is temperate semi-arid with a mean annual precipitation of 492 mm, and a mean annual temperature of 14.8 °C. The dairy farm has 800 Kiwi Cross breed (Holland × Jersey) cows under production, milking them twice a day. Herd feeding is pasture-based while maize and soybean meal is offered during milking activities. Winter supplementation is provided with maize silage, wheat bran, and soybean hulls. In the yearly average, the diet consists of 45% pasture (9 kg), 22% maize silage (4 kg), and 33% concentrate (6 kg of maize and soybean meal) per cow per day. During milk extraction animals are kept in the waiting yard (11 m-diameter area), with room/space for 400 cows. The milking yard is manually washed twice a day, flushing water on the floor with a hose. Effluents are collected in a series of 3 lagoons, the first of which is a 4 m depth anaerobic pond, sequentially connected to two 1.5 m depth aerobic ponds. Sediments are mechanically removed from the first pond once a threshold level is reached.

### 2.2. Soil and Amendments Characterization

About 5 kg of surface soil (0–20 cm depth) was collected with a shovel from an experimental plot located in Estación Experimental Agropecuaria (EEA) Hilario Ascasubi (39°23′S, 62°37′W), INTA, Buenos Aires Province, Argentina. Soils are classified as Entic Hapludoll [20], within the "La Merced" soil series, a sandy soil typical of the area [21]. The soil was air-dried and sieved (<2 mm) for chemical analyses (Table 1). Electric conductivity (EC) and pH were measured in soil:water slurries (1:2.5) by potentiometry. Organic matter (OM) was determined with the Walkley–Black method [22]; total Kjeldahl nitrogen (TKN), ammonium ($NH_4$-N) and nitrate ($NO_3$-N) forms were all determined by semi-micro-Kjeldahl [23], while extractable phosphorus (Pe) was measured by colorimetry following Bray I extraction [24]. To measure Total P, a soil aliquot was digested with hot perchloric acid followed by quantification with a high-resolution multi-type ICP emission spectrometer (ICP-AES, Shimadzu 9000) [25].

**Table 1.** General chemical and physical characterization of soil and amendments. Carbon content, measured N forms of total Khjeldal (TKN), nitrate ($NO_3$), ammonia ($NH_4$), and calculated N forms of total inorganic N (TIN), total nitrogen (TN), and organic N (No), along with total P (P) supplied by each amendment, and extractable P in the soil (Pe). All these properties are expressed on a dry weight basis ($g\ kg^{-1}$). Additional determinations include pH and EC ($dS\ m^{-1}$) for soil and amendments, and the percentages of volatile solids (VS) and total solids (TS) for each of the amendments used in the study.

| Chemical Property | Units | Soil | LDE [1] | SDE | DED | OCMC | OCMD |
|---|---|---|---|---|---|---|---|
| C | $g\ kg^{-1}$ | 14.10 | 218.25 | 241.86 | 170.46 | 213.67 | 314.89 |
| TKN | $g\ kg^{-1}$ | 1.40 | 26.34 | 10.69 | 66.31 | 15.13 | 66.56 |
| $NO_3$ | | 0.03 | 0.16 | 0.01 | 0.71 | 2.88 | 1.07 |
| $NH_4$ | | 0.01 | 9.07 | 0.28 | 47.33 | 0.08 | 52.64 |
| TIN | | 0.04 | 9.23 | 0.29 | 48.04 | 2.96 | 53.71 |
| TN | | 1.43 | 26.50 | 10.71 | 67.02 | 18.01 | 67.64 |
| No | | - | 17.27 | 10.42 | 18.98 | 15.05 | 13.92 |
| P | $g\ kg^{-1}$ | 0.42 | 6.17 | 2.87 | 5.29 | 12.72 | 8.68 |
| Pe | | 0.01 | - | - | - | - | - |
| pH | | 7.20 | 7.35 | 7.90 | 7.22 | 8.61 | 7.60 |
| EC | $dS\ m^{-1}$ | 0.10 | 10.27 | 3.40 | 12.07 | 3.03 | 12.80 |
| VS | % | - | 1.93 | 26.31 | 1.03 | 50.83 | 3.52 |
| TS | % | - | 41.05 | 41.66 | 29.69 | 33.56 | 61.52 |

[1] Liquid fraction of dairy effluent; SDE: solid fraction of dairy effluent; DED: dairy effluent digestate; OCMC: onion–cattle manure compost and OCMD: onion–cattle manure digestate.

Five amendments (raw and processed cattle manure) were produced from the manure and slurries generated in the dairy farm. The liquid fraction of the dairy effluent (LDE) and the solid fraction of the dairy effluent (SDE) were collected from the aerobic pond and the sedimentation chamber of the milking yard, respectively. The onion residues were collected from a nearby onion packing warehouse from a refuse onion pile (quality nonacceptable for export). This is mostly composed of external cataphylls, damaged bulbs, oversized or small bulbs, and bulbs affected by fungal or bacterial diseases. The feasibility and convenience of treating these locally produced, slow-degrading onion wastes through composting and co-digestion was reported previously by Iocoli et al. [6]. Thus, the onion residue was either composted with manure (OCMC) or processed through co-digestion with manure (OCMD). The anaerobically processed materials, namely, the dairy effluent digestate (DED) and the OCMD, were obtained in experimental batch digesters as reported by Iocoli et al. [6], while the aerobically processed OCMC was produced with a windrow composting system, as reported by Orden et al. [26].

Chemical determinations of EC, pH, TKN, $NO_3$, $NH_4$, and total P for each amendment (Table 1) were conducted similarly to the procedures described previously for the soil used in the incubations, with the exemption of the pH and EC, which were run without dilution for the liquid samples and with a 1:10 sample:water mass ratio for the solid samples. The additional calculations shown in Table 1 refer to total inorganic nitrogen (TIN), the sum of $NO_3$ and $NH_4$, total nitrogen (TN), the sum of NTK and $NO_3$, and organic N (No), obtained as the difference between NTK and $NH_4$.

The total C of the amendments was determined by dry combustion at 1500 °C with a LECO C Analyzer (LECO Corporation, St. Joseph, MI, USA). To prepare the liquid amendments (LDE, DED, OCMD) for this determination, a 4 $cm^3$ aliquot of each sample was added to the LECO inert absorbent material and oven-dried at 40 °C, following manufacturer recommendations. For each amendment, the total solids (TS) were determined by drying samples at 105 °C to a constant weight; the ash content was determined by burning the samples at 550 °C for 2 h, whereas the volatile solids (VS) were calculated by subtracting the ash percentages from the TS percentages.

A rapid characterization of the amendments was conducted with UV–visible and infrared (IR) spectroscopy (Supplementary Materials Table S1 and Figures S1 and S2) following a method developed by Iocoli et al. [12]. Briefly, the solid amendments were prepared as 1% Merck Uvasol potassium bromide tablets (1.8 mg dry sample in 180 mg KBr), and the liquid amendments were also processed as pellets, which were obtained by incorporating 0.30 $cm^3$ into 180 mg KBr, to achieve a dry base concentration of 0.5–1.0%. The liquid amendments were subjected to a UV–visible spectroscopic scanning between 180 and 665 nm (UV–Vis spectrophotometer PG instruments T60, UK); the IR spectra for all amendments were obtained within the mid IR range (4000–400 $cm^{-1}$) with 64 scans and an 8 $cm^{-1}$ resolution (Nicolet iS50 FT-IR Thermo Fisher Scientific, CO, Waltham, MA, USA).

The UV–visible characterization of the liquid amendments showed better absorption in the 180–320 nm (UV–Vis) range for OCMD compared to that of LDE and DED, reflecting the higher contents of aromatic, condensed aromatic, conjugated aromatic (with electro-attracting chromophores (C = C, C = O)), substituted aromatic, and highly conjugated olefinic species (Figure S1). The IR spectra of the amendments (Figure S2) showed similar absorption areas; the main chemical groups identified in the different regions are detailed in Table S1. In line with the chemical characterization, the IR spectra inspection showed a clear separation among the solid (SDE, OCMC) and the liquid amendments (LDE, DED and OCMD). In general, SDE contains a high proportion of aliphatic amides, amines, and low-molecular weight unsaturated compounds, whereas OCMC has a high proportion of heavier compounds and aromatic structures. For the liquid amendments on the other hand, DED showed high-molecular weight aliphatic compounds, polysaccharides, esters, hydroxylated compounds, aldehydes, ketones, and high-molecular weight carboxylic acids and aromatic compounds. OCMD had a high degree of intramolecular association and content of condensed aromatic compounds, along with low-molecular weight N compounds, short-chain organic acids, and free ammonium. The liquid fraction of the dairy effluent (LDE) showed soluble forms of organic matter, while the solid fraction SDE was composed of more recalcitrant forms of organic matter.

### 2.3. Soil Incubations

### 2.3.1. Microcosms Preparation

To prepare each microcosm, 100 g of air-dried sieved soil was placed in a 750 $cm^3$ glass flask and tap water was added to bring the soil to 50% of its water-holding capacity (WHC). The microcosms were pre-incubated at 25 °C in the dark for a week before adding the amendments. The amendments were supplied at an equivalent dose of 77.4 mg $kg^{-1}$ of TKN, which is comparable to the typical fertilization rate of 185 $kg^{-1}$ N ha for the onion crop in the region. Table 2 shows the amounts of C, N forms, and P supplied by each amendment on a dry weight basis at this equivalent level of TKN. The amendments were thoroughly mixed with the soil while the control soil flasks were left un-fertilized, and their moisture was corrected to 60% of WHC. A total of 180 microcosms were set in a completely randomized design, with three replicates per each of the six treatments (five amendments plus control). The moisture content was kept constant throughout the incubation by weighing the microcosms every other day and adding water gravimetrically to correct the weight losses. Incubations were done at 25 °C in the dark as for the pre-incubation stage.

**Table 2.** Carbon content, measured N forms of total Khjeldal (TKN), nitrate ($NO_3$), ammonia ($NH_4$), and calculated N forms of total inorganic N (TIN), total nitrogen (TN), and organic N (No), along with total P (P) supplied by each amendment in the study. All properties are expressed on a dry weight basis (mg $kg^{-1}$) at the equivalent level of TKN.

| Chemical Property | Units | LDE [1] | SDE | DED | OCMC | OCMD |
|---|---|---|---|---|---|---|
| C | mg $kg^{-1}$ | 641.1 | 1750.4 | 198.9 | 1092.7 | 366.1 |
| TKN | mg $kg^{-1}$ | 77.4 | 77.4 | 77.4 | 77.4 | 77.4 |
| $NO_3$ | | 0.5 | 0.1 | 0.8 | 14.7 | 1.2 |
| $NH_4$ | | 26.7 | 2 | 55.2 | 0.4 | 61.2 |
| TIN | | 27.1 | 2.1 | 56.1 | 15.1 | 62.4 |
| TN | | 77.9 | 77.5 | 78.2 | 92.1 | 78.6 |
| No | | 50.7 | 75.4 | 22.1 | 77 | 16.2 |
| P | mg $kg^{-1}$ | 18.1 | 20.8 | 6.2 | 65 | 10.1 |

[1] Liquid fraction of dairy effluent; SDE: solid fraction of dairy effluent; DED: dairy effluent digestate; OCMC: onion–cattle manure compost; OCMD: onion–cattle manure digestate.

### 2.3.2. C Mineralization from Microcosms

Of the 180 microcosms, 18 were allocated to static incubation to assess carbon mineralization by measuring the release of $CO_2$ [27] over a period of 119 days (17 weeks) comparable to the growing season for onions in the region, from transplant (October) to harvest (February). Briefly, a beaker with 25 $cm^3$ of NaOH trap solution (0.5 M) was placed inside each flask. Traps were removed and replaced 8 times during the incubation (3, 7, 21, 35, 49, 70, 91 and 119 days after amendment), and at each time, the amount of $CO_2$ in the flasks was determined by back-titration of the trap solution with HCl (0.25 M) in an excess of $BaCl_2$-saturated solution (1.5 M).

The $CO_2$-C was calculated by multiplying the $CO_2$ production at each time by the coefficient 0.273 (C proportion in the $CO_2$ molecule). The apparent mineralization of the organic C (%Cam) from each of the amendments was calculated as the difference between the cumulative $CO_2$-C evolved in the amended soil and the cumulative $CO_2$-C produced in the un-amended soil (control) at the end of the incubation period. This difference was in turn expressed as the proportion of the C added with each of the amendments [28] (1):

$$\%\text{Cam} = (\text{CumCO}_2\text{-C}_{\text{treatment}} - \text{CumCO}_2\text{-C}_{\text{control}})/\text{C}_{\text{added}} \times 100 \tag{1}$$

### 2.3.3. N and P Mineralization from Microcosms

The remaining 162 microcosms were used in a parallel incubation study to assess N and P mineralization, destructively sampling 3 replicates at each of 9 sampling times, for each of the six treatments under comparison. At each sampling date, moisture content, $NO_3$, $NH_4$, and Pe were determined in the amended and unamended soils, checking moisture every 2–3 days and restoring it to 60% WHC when needed. Likewise, to monitor changes in soil conditions, soil pH and EC were determined at each sampling time via potentiometry, as previously described (Table S2 and Figure S3).

The additional calculations used in the results and the discussion section include the following.

Net available N (NaN), calculated for each date with the following Formula (2):

$$\text{NaN} = \text{TIN(t)}_{\text{treatment}} - \text{average TIN(t)}_{\text{control}} \tag{2}$$

where t is time (sampling day) (t $\geq$ 0).

The net mineralized N (NmN) (3):

$$\%\text{NmN} = [\text{TIN(t)} - \text{average TIN(t = 0)}_{\text{treatment}}] - [\text{average TIN(t)} - \text{TIN (t = 0)}_{\text{control}}]/\text{No} \times 100 \tag{3}$$

Net extractable P was calculated from Pe, according to the Formula (4):

$$NPe = Pe(t)_{treatment} - \text{average } Pe(t)_{control} \tag{4}$$

Finally, the efficiency of amendments in supplying available P as a percentage of the added P source was determined as (5):

$$\%Peff = Pe_{treatment} (t = 119) - \text{average } Pe_{control} (t = 119)/P_{added} \times 100 \tag{5}$$

### 2.4. Statistical Analyses

As previously stated, incubation studies were set up in a completely randomized design with three replications at each of the nine times. Linear models were fit to each studied variable of C mineralization ($CO_2$-C release rate, cumulative $CO_2$ release, %Cam), N mineralization (TIN, NaN, %NmN) and P mineralization (Pe, NPe, %Peff), as well as supplementary parameters (pH and EC), using the PROC GLIMMIX of SAS software version 9.4 (SAS Institute, Cary, NC), and using treatment as a fixed effect in the analyses of variance. Because C mineralization was measured at successive times on the same experimental units, it was analyzed using a repeated measures approach with a heterogeneous autoregressive [ARH(1)] model for the variance–covariance matrix of the residuals [29]. The least square means were separated using the PDIFF option of LSMEANS in PROC GLIMMIX; the least significant differences (LSD) at the $\alpha$ level ($\alpha$) = 0.05 are reported in plots and tables to allow readers to easily make mean comparisons among treatments. The LSD value was computed by multiplying the appropriate t value by the standard error of the difference of means (SED) provided in the output from the PDIFF option of the LSMEANS statement. In the figures, the LSD values have been plotted as a vertical bar for each time to facilitate visual comparisons among treatments (e.g., if the vertical difference between two treatments is bigger than the LSD bar span, treatments are statistically different). To complement this visual representation, a table of mean values for each studied variable and their corresponding LSDs is included as Supplementary Materials. The plots were created using Sigma Plot 12.5 (Systat Software, Inc., San Jose, CA, USA).

### 3. Results

#### 3.1. C Mineralization

The results from the repeated measures ANOVA on $CO_2$ released show a statistically significant treatment × time interaction effect (F = 72; $p < 0.0001$), indicating a varying response to the addition of amendments measured at successive times during the 17 weeks of the incubation period (Figure 1a, Table S3).

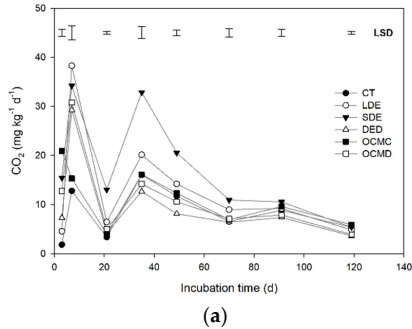

(a)

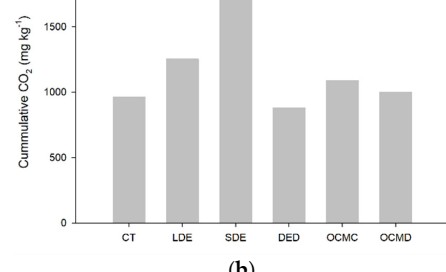

(b)

**Figure 1.** *Cont.*

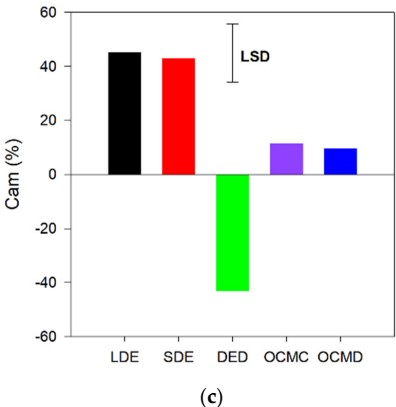

(c)

**Figure 1.** (**a**) Daily $CO_2$ released (mg kg$^{-1}$) by untreated controls (CT) and soils incubated with LDE, liquid fraction of dairy effluent; SDE: solid fraction of dairy effluent; DED: dairy effluent digestate; OCMC: onion−cattle manure compost; OCMD: onion−cattle manure digestate, over a 119−day (17 weeks) incubation period; (**b**) cumulative $CO_2$ released (mg kg$^{-1}$) by the treatments at the end of the incubation period; (**c**) apparent mineralization of the organic C (Cam, %) estimated for each of the amendments based on the initial C amount supplied and the cumulative release of $CO_2$ at the end of the incubation period. In all panels, the LSD bar represents the least significant difference value (alpha = 0.05) to facilitate visual comparison among treatment means.

Thus, at three days from the start of the incubation, OCMC generated the most $CO_2$, followed by SDE and OCMD, whereas DED and LDE had lower levels of emissions, yet still higher than the controls. While the $CO_2$ emissions from most amendments and the controls peaked at seven days, the $CO_2$ released from the OCMC decreased at that sampling time, reaching its minimum value at 21 days (3 weeks). At this early sampling time, LDE had the highest $CO_2$ emission followed by SDE and the digestates DED and OCMD, which had similar levels of $CO_2$ emissions. From day 21 to day 70 (10 weeks), SDE showed the highest $CO_2$ released compared to all the other amendments, yet during the 91-day (13 weeks) sampling time, the emissions of the SDE treatment were not statistically different from the emissions of soils with LDE and OCMD.

A second peak for all amendments occurred at 49 days of incubation, where the emissions from LDE were statistically different (lower) from those described for SDE. The $CO_2$ emissions were even lower for OCMC, but were not different from the controls, yet they differed from those of the digestates DED and OCMD. The release of $CO_2$ from these digestates remained lower than all other treatments, including the unamended controls, until the end of the incubation period.

Accordingly, the cumulative $CO_2$ released at the end of the 17 weeks (Figure 1b) showed a statistically significant effect of the treatments (F = 81; $p < 0.0001$). SDE released the most cumulative $CO_2$ and it was statistically different from all other treatments. The second highest emissions were recorded for the LDE, and this was also statistically different from the remaining treatments. The OCMC was not statistically different from OCMD, yet it was found to be statistically different from both the control and the DED. Lastly, both digestates OCMD and DED, though statistically different in their respective cumulative emissions, did not differ from the cumulative $CO_2$ released by the unamended controls (CT).

These results are consistent with the results from the ANOVA on the apparent mineralizable C (Cam) available from the amendments, which showed a statistically significant effect of treatments (F = 21; $p < 0.0001$) (Figure 1c). Three groups were distinguishable regarding Cam; while there was a negative measure for DED (−43%), Cam was the highest for LDE and SDE treatments, whereas a much lower yet positive Cam was available with OCMC and OCMD.

### 3.2. N Mineralization

The results from the two-way ANOVA on TIN show a statistically significant treatment × time interaction effect (F = 9; $p < 0.0001$), indicating a varying response to the addition of amendments measured at successive times over the 119-day incubation period (Figure 2a, Table S4). Thus, at the start of the incubation, the TIN content was highest in soils amended with both digestates (OCMD and DED), and lowest in the SDE-amended soil and CT, whereas LDE and OCMC showed intermediate levels between those two groups. The initial TIN content was also statistically different between LDE and OCMC. The digestates (OCMD and DED) had the highest levels of TIN for 70 days, yet OCMD showed higher TIN than DED by the end of incubation at 91 and 119 days. On the contrary, CT and SDE had the lowest levels of TIN for the first 3 weeks from the start of the incubation, whereas SDE showed lower TIN than CT from day 35 to day 70, and again at day 119. The LDE and OCMC treatments had intermediate levels of TIN, while LDE had significantly higher levels than OCMC on days 0, 7, 49 and 70, and the compost showed higher TIN than LDE at days 91 and 119.

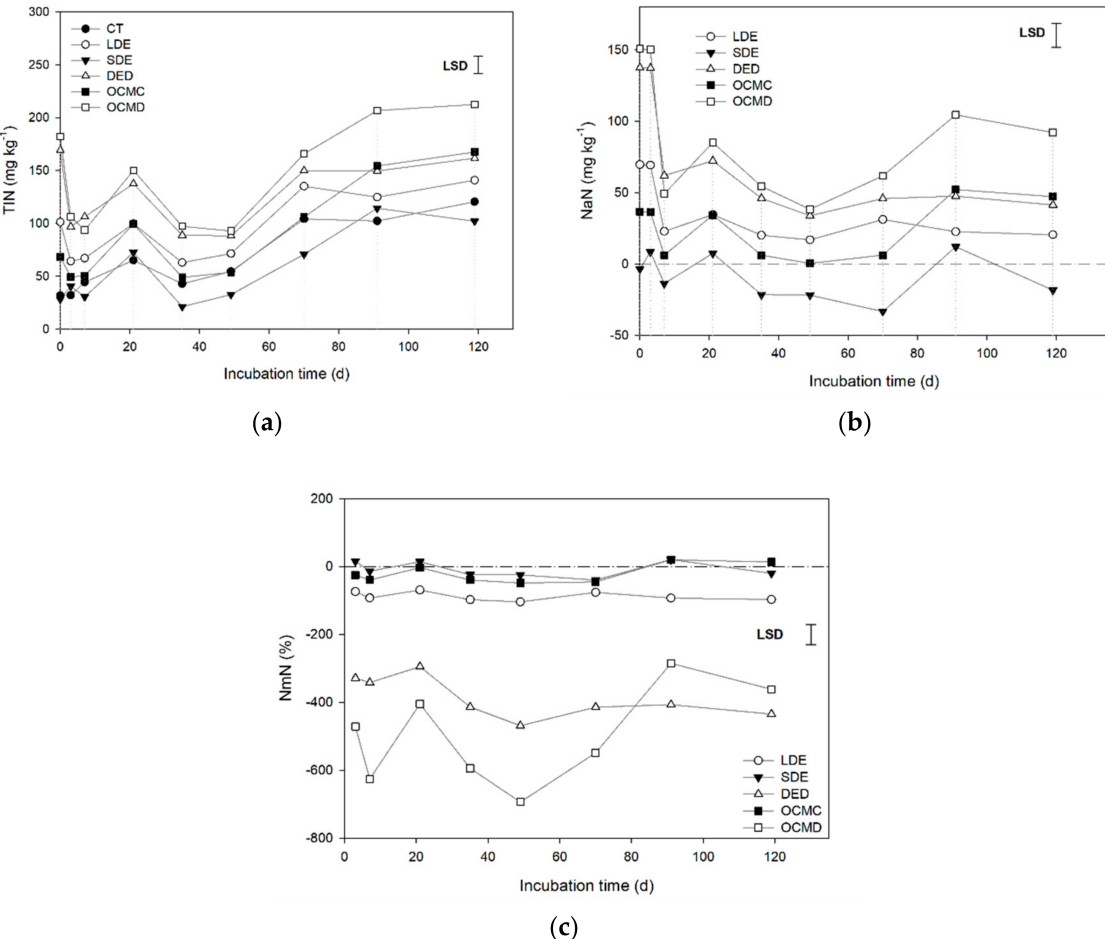

(a)

(b)

(c)

**Figure 2.** (**a**) Total inorganic nitrogen (TIN, mg kg$^{-1}$) calculated as the sum of nitrate plus ammonium (NO$_3$ + NH$_4$) for untreated controls (CT) and soils incubated with LDE, liquid fraction of dairy effluent; SDE: solid fraction of dairy effluent; DED: dairy effluent digestate; OCMC: onion−cattle manure compost; OCMD: onion−cattle manure digestate, over a 119−day incubation period; (**b**) net available N (NaN, in mg kg$^{-1}$) calculated as the difference between TIN in each treatment and the CT at each sampling time; (**c**) net mineralizable N (NmN, %) calculated as the proportion of the organic N added that is available for mineralization at each sampling time. In all panels, the LSD bar represents the least significant difference value (alpha = 0.05) to facilitate visual comparisons among treatment means.

Within the first week after the start of incubation, the TIN content declined in most treatments, but not in the CT and SDE, and subsequently peaked at day 21 in all treatments and CT. After this peak in TIN, there was a period between days 35 and 49 that showed constant TIN levels in all treatments, followed by an increase on day 70 that lasted until the end of the incubation for OCMD, OCMC, and SDE, but again stabilized for the CT, LDE, and DED.

Consistent with the TIN results, the results from the ANOVA on NaN showed a statistically significant treatment $\times$ time interaction effect (F = 8.31 $p < 0.0001$), indicating a varying response to the addition of amendments measured at successive times (Figure 2b, Table S4). From the beginning of the incubation and until day 70, NaN was highest in soils amended with both digestates (OCMD and DED), and lowest in the SDE amended soil, whereas LDE and OCMC showed intermediate levels, yet there were significant differences between them during the first week and on day 70. Late in the incubation, at days 91 and 119, the highest NaN value was still obtained in the OCMD treatment but was significantly different from DED, while this was similar to the compost (OCMC). The SDE treatment still showed the lowest NaN level, yet it did not differ from LDE on day 91.

Seven days after the start of incubation, the NaN levels declined in all treatments, and subsequently peaked at day 21 in OCMC and OCMD, while remaining constant in DED and LDE throughout the incubation. After this peak in NaN, there was a period between days 35 and 70 that showed constant levels in both treatments, followed by an increase on day 91 that lasted until the end of the incubation. Although SDE showed similar temporal dynamics to OCMD and OCMC, the NaN values were positive only at days 3, 21 and 91.

The results from the ANOVA on %NmN showed a statistically significant treatment $\times$ time interaction effect (F = 9.1; $p < 0.0001$), indicating a varying response to the addition of amendments measured at successive times (Figure 2c, Table S4). The percent NmN was highest in SDE and OCMC throughout the 17 weeks of incubation, yet at days 3 and 21, the OCMC values did not differ from those of LDE, and this one's values were not statistically different from both SDE and OCMC at day 70. In contrast, OCMD showed the lowest %NmN values from the start of the incubation until day 70, yet at days 91 and 119, DED exhibited the lowest %NmN. It is noteworthy that only SDE and OCMC showed positive %NmN values occasionally during the incubation. The other treatments always rendered negative values, yet LDE showed higher values than OCMD and DED, and this in turn had statistically significantly higher %NmN values than OCMD during most of the incubation period.

The %NmN levels remained constant throughout the incubation in the SDE, OCMC and LDE treatments, yet in both solid amendments there was a transient increase at day 91. In contrast, inspection of Figure 2c and Table S4 shows that both digestates had similar temporal dynamics, yet DED peaked at day 21 and then declined to lower, more constant %NmN levels than at the first week of incubation, while OCMD had two peaks at 21 and 91 days after the start of incubation.

### 3.3. P Mineralization

The results from the ANOVA on Pe showed a statistically significant treatment $\times$ time interaction effect (F = 22; $p < 0.0001$), indicating a varying response to the addition of amendments measured at successive times (Figure 3a, Table S4). At the start of incubation (day 0), all treatments and the control were significantly different from each other, except for the Pe contents in DED and OCMD, which were not different between them. The highest Pe content was measured in the OCMC soils, while the CT showed the least throughout the incubation. Three days after the initiation of the incubation, Pe levels declined in all treatments and the CT, although this reduction was most noticeable in the OCMC, LDE and SDE treatments. At this early sampling point, all treatments differed from the CT. Moreover, the OCMC and LDE treatments differed from each other and from the rest of the treatments, while SDE and the digestates (OCMD and DED) showed no differences among them. One week after the start of the incubation, most treatments differed from

the CT and among themselves, except for DED, which had similar Pe levels to the CT and OCMD, and the latter, which did not differ from SDE. At the third week of incubation, all treatments differed from the CT and among themselves, except for the digestates (DED and OCMD) and SDE, which had similar Pe levels. From day 35 to day 119, most treatments differed from the CT and among themselves, except for DED treatment, which did not differ from CT, and OCMD, which showed no difference from SDE. Yet the digestates (DED vs. OCMD) were not statistically different on day 49, and SDE differed from OCMD on day 91.

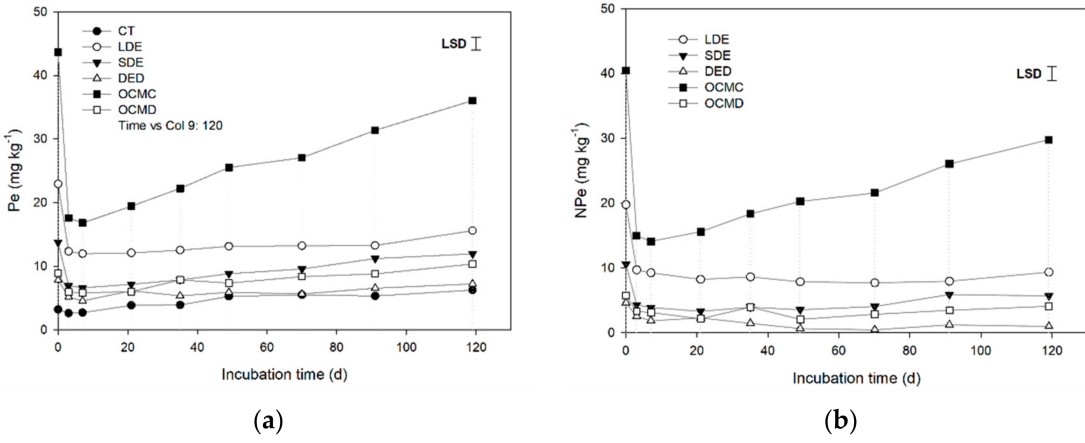

(a)                                           (b)

**Figure 3.** (**a**) Extractable P (Pe, mg kg$^{-1}$) measured in untreated controls (CT) or soils incubated with LDE, liquid fraction of dairy effluent; SDE: solid fraction of dairy effluent; DED: dairy effluent digestate; OCMC: onion–cattle manure compost; OCMD: onion–cattle manure digestate, over the 119-day incubation period; (**b**) net extractable P (mg kg$^{-1}$) calculated as the difference between Pe in each of the amendments and the Pe in the untreated CT, at each sampling day, during the 119-day incubation period. In all panels, the LSD bar represents the least significant difference value (alpha = 0.05) used to facilitate the visual comparison among treatment means.

The Pe level remained relatively constant in the CT and all treatments except for OCMC until day 35 of incubation, when the Pe level of CT started to increase gradually until the end of incubation. The level of Pe recovered more slowly in the SDE and OCMD treatments, which showed slightly increasing levels from day 49 until the end of the incubation. The soils amended with DED and LDE showed a slight increase in the Pe level later during the incubation, in the last sampling days (91 and 119 days). The OCMC treatment showed a contrasting temporal dynamic for Pe level with respect to the rest of the treatments and CT, as the Pe level increased steadily after the initial drop and until the end of the incubation.

Accordingly, the results from the ANOVA on NPe showed a statistically significant treatment × time interaction effect (F = 20; $p < 0.0001$), indicating a varying response to the addition of amendments measured at successive times (Figure 3b, Table S5). The NPe temporal dynamics were similar to the dynamics observed for Pe, with the highest NPe values for each treatment obtained at the start of the incubation, followed by a drop during the first week, and displaying an almost constant NPe value until the end of the experiment for most treatments except the compost (OCMC). Over the entire incubation period, the highest NPe was measured with the OCMC, while DED yielded the lowest NPe values. At the start of the incubation (day 0), most treatments differed from each other, except for the digestates (DED and OCMD), which showed no statistically significant difference among themselves. From day 3 to day 21, the OCMC and LDE treatments differed between each other, and from the digestates and SDE, yet there were no statistical differences among the members of this latter group of amendments. From day 35 until the end of the incubation (17 weeks), most treatments differed from each other, except for SDE and OCMD, which were not different from each other in their NPe release. In addition, the OCMD treatment was similar to the other digestate (DED) at day 35.

The results from the ANOVA on the efficiency of supplying available P (%Peff) from the amendments show the statistically significant effect of the treatments (F = 43.6; $p < 0.0001$) (Figure 4, Table S5). Two groups were distinguishable regarding %Peff: three treatments had values higher than 40%, while two treatments showed considerably lower values (<30%). The highest %Peff was obtained with LDE (51%), and was significantly higher than %Peff for OCMD, although it was not different from OCMC. The latter amendment, in turn, did not differ from OCMD. The lowest %Peff was obtained with DED (15%); moreover, DED and SDE treatments were significantly different between themselves and from the rest of the treatments as well.

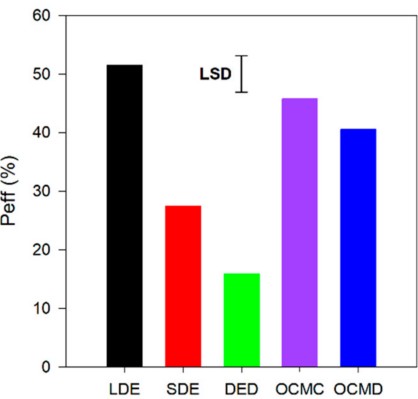

**Figure 4.** Efficiency of supplying available P (%Peff) estimated for each of the amendments based on the residual Pe at the end of the incubation and the initial P amount supplied with the amendments. The LSD bar shows the least significant difference value (alpha = 0.05) used to facilitate the visual comparison among treatment means.

## 4. Discussion

### 4.1. Amendment Characterization and Its Implications

The amendments from the solid and liquid fractions of the dairy farm effluents (SDE and LDE) were highly dissimilar. The LDE showed higher C and lower TN contents (i.e., lower C/N ratio), as well as a higher TIN, in agreement with the similar findings for liquid dairy effluents reported in New Zealand [30]. Spectroscopic characterization also suggests that it has a low proportion of polycondensed aromatic compounds, and is composed of soluble sugars and amino sugars, amino acids, proteins and short-chain organic acids [31] that may be rapidly degraded in soil. In contrast, the solid fraction of the dairy effluent (SDE) has a high C/N ratio and very low TIN content, with most N in the organic form, in coincidence with results reported by Longhurst et al. [30]. The SDE is mainly composed of fibers (cellulose, hemicellulose and lignin), and thus, is decomposed more slowly than the liquid effluent [32]. The composting of manure, as in the OCMC treatment, results in big losses of C as gaseous $CO_2$ through microbial respiration, and the reduction in $NH_4$ content through immobilization and nitrification, while $NO_3$ loads are concomitantly increased [33] and potentially lost via denitrification processes or leaching [34]. As a result of the bigger losses of C in comparison to N reduction, together with the constant P level, the OCMC had an excess of P in relation to N. A ratio of $NH_4$ to $NO_3 < 1$, as well as the high C and low TN (i.e., high C/N ratio), in the OCMC reflects an appropriate stability condition [35,36]. The anaerobic digestates (DED and OCMD) were considerably similar in their chemical composition. Both digestates were rich in aromatic compounds and $NH_4^+$, based on their spectroscopic and chemical characterization. The DED showed a decrease in the C content and increases in TN and No fractions, which reduced both C/N and C/No ratios compared to LDE, reflecting the recalcitrant nature of the remaining No. In the anaerobic digestion process, labile organic matter is transformed to $CH_4$ and $CO_2$, while recalcitrant compounds accumulate [37,38], and the inorganic elements (N, P, K) are



preserved in the digestate [39]. As in this study, other researchers working with digestates from different residues confirmed that the most prevalent N form is $NH_4$-N [40,41].

The pH values decreased slightly over the incubation period (Table S2) as a result of the nitrification of $NH_4$ incorporated with the amendments and/or from the mineralization of amendments and native SOM [42]. The OCMC showed the least variations in pH, likely due to a limited supply of $NH_4$ (Table S2), and a higher OM stability and buffering capacity when applied to soil [43]. In contrast, OCMD showed the strongest reductions in soil pH, probably due to the large input of $NH_4$ in the amendment. From an agronomic perspective, none of the analyzed treatments produced pH changes that would mean a problem for crop development.

### 4.2. C Mineralization

The marked differences in composition and characteristics among amendments reveal their contrasting effects on C mineralization when applied to soils. Digestates may be used as fertilizers given their rapid supply of available nutrients and the improvement of soil quality in the long term. On the other hand, composts are good soil conditioners due to their supply of stable OM, while the supply of nutrients through mineralization is usually low [44,45]

Anaerobic digestion is a highly efficient way of reducing the OM content of residues when correctly managed [39]. In this study we have carefully set the retention time of residues in anaerobic digesters to warrant the degradation of labile OM and increase the level of recalcitrant OM. The stability of the digestates is reflected by the C mineralization dynamics, which showed a rapid initial mineralization followed by a period of net C immobilization. Béghin-Tanneau et al. [46] confirmed a negative priming effect post-application of digested $^{13}$C-labeled maize silage, in line with our results. With regard to compost, the OCMC used in this study was highly stable, considering the criteria of respiration rates < 1 mg $g^{-1}$ $d^{-1}$ of $CO_2$-C that Wang et al. [47] set to define a highly stabilized compost from cattle and pig manures.

The use of digestates as soil amendment elicited a rapid initial release of $CO_2$ from microbial respiration, yet the cumulative $CO_2$ released and the elicited respiration dynamics were similar between them and with the unamended controls. The initial burst in biological activity may be attributed to the high input of labile short-chain organic acids [28,48,49]. Kirchmann and Lundvall [40] also observed rapid degradation (within 1 day) of short-chain organic acid content in pig slurry, pig slurry digestate, and cattle manure digestate, in agreement with our results.

The cumulative respiration of soils with unprocessed amendments (LDE and SDE) was higher than the cumulative $CO_2$ measure in soils with processed materials (DED, OCMD, and OCMC). The addition of LDE to soils led to the strongest initial stimulation of microbial activity, likely due to a greater proportion of labile compounds within this unprocessed amendment. Compared with the response observed for LDE, SDE-amended soils showed a slower increase in microbial respiration, yet greater cumulative $CO_2$ respiration, likely associated with the presence of more complex organic matter in this type of amendment. These results agree with those reported by Busby et al. [50]. It is noteworthy that the unprocessed amendments had the highest apparent mineralization of organic C (Cam) due to the higher proportion of degradable organic compounds added and their consumption during the incubation period when compared to the other amendments [51,52]. The negative Cam measured for DED evidences their lower C content when compared to the OCMD digestate, and the presence of recalcitrant materials not available for microbial use resulting from the anaerobic digestion process [37,38].

### 4.3. N Mineralization

The levels of TIN were initially high upon addition of amendments to soil, but then rapidly reduced (within 1 week of incubation), in most treatments except in SDE and CT. This result may be explained by the large N demand of the microbial community [40]. Mi-

crobes preferentially use ammonium as a N source, although nitrate is also used to sustain the N requirement [53,54]. A rapid initial reduction in N occurred simultaneously with the highest $CO_2$-C emission rate, supporting our hypothesis that microbial immobilization was the main mechanism of N depletion in our study [55–58]. This is further supported by the results reported in other soil systems [28,48,59]. Nitrogen losses via volatilization were considered negligible, as amendments were thoroughly mixed with the soil at the time of application, and air flow through it was limited [59].

The high content of labile organic compounds in the anaerobic digestates may explain the fast N assimilation in the microbial biomass, in line with Kirchmann and Lundvall [40], which showed a significant correlation between the fatty acid content of the digestate and the N immobilization levels. Despite the suggested role of the microbial immobilization of N, the soils amended with digestates still showed levels of TIN higher than those measured under the control conditions, as also observed by de la Fuente et al. [59]. The use of digestates as soil amendments produced the largest increase in the TIN soil content within 3 weeks of incubation, and again four weeks later, yet the OCMD amendment showed the highest TIN by the end of the incubation, in agreement with previous reports [48,60]. Compost and LDE application moderately increased the TIN content in soil at the time of amendment, and rapidly returned to basal values similar to or slightly higher than the CT, as reported by others [61,62]. The N immobilization after amendment may be due to the low content of N compounds in composts [63]. The TIN level in LDE-amended soil showed similar temporal dynamics to OCMC-treated soil, due to the similar C/N and C/No ratios among them, with most N under the organic form (Table 1). As for the SDE amendment, it released less TIN than the CT soil during most of the incubation, which could be explained by its high proportion of recalcitrant OM and organic N slowly releasing available inorganic N [30], which could be rapidly immobilized in the microbial biomass.

While the net available N (NaN) results showed similar temporal dynamics to the TIN content, this variable is especially useful for analyzing the behavior of the amendments with regards to providing available N to the crops. Solid amendments were less prone to release inorganic N in levels higher than soil basal levels. While OCMC showed a low content of NaN ($<10$ mg kg$^{-1}$) for almost 7 weeks (except on day 21), and this increased thereafter, SDE had negative NaN values during most of the incubation time, which were likely attributed to the microbial immobilization of soil N [34]. These results also agree with those of Longhurst et al. [30], who reported that the very low proportion ($<5\%$) of mineral N in the solid amendments could produce a retarded N release for plants. In contrast, the liquid amendments had consistently higher ($>10$ mg kg$^{-1}$) positive NaN values. The labile nature of organic N compounds (mostly urea) present in LDE means that it can be easily transformed into TIN, being susceptible to losses through leaching and denitrification [34]. Therefore, liquid dairy effluents and digestates should be carefully managed and applied in a timely manner in order to prevent TIN losses.

The NmN variable, in turn, reflects the release of available N as a proportion of the organic N added with each of the amendments that is mineralized at each sampling time. Dosification of amendments was done based on a fixed TKN level (77.4 mg kg$^{-1}$ soil). As a result, the digestates supplied the lowest No amounts, while unprocessed amendments (SDE, LDE) and compost supplied relatively higher amounts. There were two clear patterns in the NmN response: while digestates showed the lowest percentage of mineralization throughout the incubation, the unprocessed amendments and compost showed higher yet mostly negative and relatively stable %NmN values. Negative NmN values mean that N immobilization rather than N release is the predominant fate of mineralized N in the soil. Compost amendment to soil is an effective strategy to provide a slow-release source of nitrogen for plants [61]. The similar behaviors of OCMC and SDE may be explained by the fact that SDE underwent a period of settling and storage in the sedimentation chamber, losing N through similar processes to compost and retaining more recalcitrant organic N. With regard to digestates, the relatively low content of No available for mineralization together with its recalcitrance resulted in a strong immobilization. As the most dynamic

pool of OM in the soil, immobilization by the microbial biomass reduces the risk of N losses while keeping N relatively available for plants. In agreement, other studies have reported that the use of digestates as fertilizers produced crop growth comparable to that obtained with inorganic fertilizers [6,28].

### 4.4. P Mineralization

Our results suggest that the P dynamics (both Pe and NPe) closely followed the C mineralization process, responding to energy acquisition instead of the biochemical (i.e., enzymatic) pathway, according to the hypothesis proposed by Oehl et al. [64]. The highest level of extractable P (Pe) in the amended soils was observed right after application [65], and it was rapidly reduced as a result of microbial assimilation, surface adsorption [66], and precipitation with calcium [19], and to a lesser extent, with iron and aluminum ions [67]. The fast decline in Pe level in compost-amended soil after 3 days of application may be explained by precipitation processes with Ca, as the pH was above 6 during the entire incubation period, and Fe/Al precipitation may be considered negligible [68]. In turn, microbial assimilation could also be excluded as a causal factor given that there were no differences in microbial activity between compost-amended soil and the unamended controls. The reduction in Pe levels in the rest of the treatments occurred in parallel with the highest rates of $CO_2$-C evolution and N immobilization, which suggests that P was immobilized by microbial assimilation. Overall, nutrient dynamics were closely linked to the supply of labile C, the main factor affecting microbial activity in agricultural soils [69].

The P supplied by digestates was only in the inorganic form, as observed for N release. The low supply of Pe with digestates may be due to the formation of struvite crystals and other inorganic forms of P that are not transformed during anaerobic digestion [9]. This, in turn, explains why the digestates induced a relative P deficit, despite having a N/P ratio appropriate for onion crop production. In contrast, SDE and compost amendments produced an increase in the Pe level relative to the control during the entire incubation period. That increase may be attributed to the mineralization of exogenous OM, as the redissolution of inorganic precipitates would have taken longer than the period under study (17 weeks) [19].

The greatest ability to supply available P with LDE and OCMC in comparison to the other treatments may be due to the higher proportion of available P and organic P compounds that may be used directly by microorganisms [70]. Composts, biosolids, and manure with a lower N/P ratio than the ratio necessary to meet the plants' requirements (<4) may lead to excess levels of P in the soil, and associated environmental risks when applied based on their N supply [58,71]. In contrast, DED and SDE had the lowest ability to supply available P, as they were composed mostly of recalcitrant OM and inorganic precipitates. DED showed the opposite effect, as it produced a relative Pe deficit. Dodd and Sharpley [72] postulated that a low P supply in parallel with excess labile C and N may increase microbial activity, promoting phosphatase production and the subsequent release of available forms of P from organic and inorganic sources, without compromising crop yields. However, in this research, we did not observe an increase in the ability to supply available P to the soil solution in the digestate treatment, characterized by high labile C and $NH_4$.

## 5. Conclusions

This study resulted from a collaboration among INTA, university personnel, and a local dairy farm, to generate baseline information to develop guidelines for the responsible management of waste management strategies and technologies for dairy farmers in the semi-arid region of the southwestern Buenos Aires Province in Argentina. Specifically, we evaluated the mineralization potential of dairy waste as unprocessed materials, and as processed forms, over a period of time relevant to onion production, the most important horticultural crop in the region. Thus, we characterized the release of $CO_2$, N, and P from raw, composted and digested manure amendments, in order to evaluate the feasibility

of their in-farm production and use as organic fertilizers. The elemental and structural differences among amendments, despite their common origin, led to contrasting profiles of C, N, and P mineralization, and thus in nutrient availability, immobilization, and $CO_2$ emission. Digestates showed a rapid initial C mineralization followed by a period of net C immobilization after soil application, which likely contributed to higher C retention in soil. Their main contribution as fertilizer sources was the supply of available N, while the supply of organic N and P was negligible due to the recalcitrant nature of the organic structures that contain these elements. As a result, the digestates increased the N/P ratio in the soil and caused a relative P deficit. The compost was highly stable and generated a small initial contribution of inorganic N, while the Pe supplied rapidly diminished, likely due to precipitation processes. During the entire incubation period, the compost amendment consistently released N and P via mineralization. The main contribution of compost as a fertilizer source was P supply, reducing N/P ratio and producing a relative deficit of N, in contrast to the digestates. Both processed materials, digestates and composts, were more stable than untreated residues, reducing C emission. Future studies should aim at evaluating fertilization strategies that combine both kinds of amendments, in order to exploit their complimentary agronomic characteristics. The availability of this information represents a first step towards developing the necessary guidelines for dairy waste management in the region, in order to help producers improve the sustainability of their operations and prevent environmental hazards.

**Supplementary Materials:** The following are available online at https://www.mdpi.com/article/10.3390/agronomy11122595/s1. Table S1. Location of principal indicator bands and the assignment to functional groups for the spectroscopic characterization of amendments displayed in Figure S2: FT-IR spectra. Figure S1: UV–visible spectra for liquid materials. Table S2: Mean values of soil pH and electrical conductivity over the incubation period. Table S3: Mean values of daily and cumulative $CO_2$ released, and apparent C mineralization estimated for each amendment. Table S4: Mean values of nitrogen forms measured and calculated for each amendment. Table S5: Mean values of extractable and net extractable P for all amendments under study. Tables S2 to S5 also include LSD values for each set of results.

**Author Contributions:** Conceptualization, G.A.I. and L.O.; methodology and resources, G.A.I., F.M.L. and L.O.; formal analysis and data curation, M.C.Z. and M.B.V.; writing—original draft preparation, M.C.Z., M.B.V. and G.A.I.; writing—review and editing, M.C.Z. and M.B.V.; visualization, M.B.V.; project administration, M.C.Z.; funding acquisition, M.C.Z., M.A.G. and M.B.V. All authors have read and agreed to the published version of the manuscript.

**Funding:** This research was funded by SGCyT, UNS, grant number PGI 24/A250, by Agencia Nacional de Promoción Científica y Tecnológica, grant number PICT 2014–1760 and by Consejo Nacional de Investigaciones Científicas y Técnicas, grant P-UE 22920160100031CO.

**Institutional Review Board Statement:** Not applicable.

**Informed Consent Statement:** Not applicable.

**Data Availability Statement:** Not applicable.

**Acknowledgments:** The authors are indebted to Olga Inés Pieroni (INQUISUR) for her collaboration on IR spectra interpretation, to Mariana Dennehy (INQUISUR) for allowing us to access her laboratory, to Miriam Crespo (LANAQUI) for her collaboration on amendments characterization, to Luciana Gisele Dunel Guerra and Romina Storniolo (EEA Ascasubi, INTA) for their collaboration in the preparation of microcosms, to Andrea Mairosser (EEA Ascasubi, INTA) for giving us the onion–cattle manure digestate, and to Ana María Zamponi for her collaboration in lab activities.

**Conflicts of Interest:** The authors declare no conflict of interest.

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
