# Peer review of "Towards Sustainable Dairy Production in Argentina: Evaluating Nutrient and CO2 Release from Raw and Processed Farm Waste"

_agronomy, doi:10.3390/agronomy11122595_

Round 1

Reviewer 1 Report

It addresses issues of economic, environmental and social importance. The need for the research is well justified in the Introduction and the purpose is clearly stated. The materials and methods are presented comprehensively. The results of the study are discussed extensively and embedded in existing knowledge. The conclusions are clear. The practical implications of the research are also given.

My only suggestion is to avoid/reduce describing in the Results chapter the content of tables and figures and referring to the statistical procedures used (after all, these are given in the relevant chapter), and to concentrate on the "pure" effects of the research.

Congratulations to the Authors!

Author Response

Response to Reviewer #1 comments:

It addresses issues of economic, environmental and social importance. The need for the research is well justified in the Introduction and the purpose is clearly stated. The materials and methods are presented comprehensively. The results of the study are discussed extensively and embedded in existing knowledge. The conclusions are clear. The practical implications of the research are also given.

We sincerely appreciate your positive comments, thank you!

My only suggestion is to avoid/reduce describing in the Results chapter the content of tables and figures and referring to the statistical procedures used (after all, these are given in the relevant chapter), and to concentrate on the "pure" effects of the research.

Thank you for this comment. We have revised the Results section and shortened it as per your suggestion. We have moved the guidelines for the visual interpretation of the LSD bar for mean comparison purposes in section 2.4. We are open to shortening the section further if you would like to provide more specific guidelines.

Congratulations to the Authors!

We are delighted by your appreciation of our hard work, thanks again.

Reviewer 2 Report

This is an interesting study, and it is easy to read.

Here some suggestions:

In the introduction put the attention not only on Argentina but extend the area at world level, at least at aera with similar condition to that one investigated in the present research, on the other hand put the word “Argentina” in the title.

Add the kg of feeding using per cow

Explain better the use of onion by-products and why in combination with the manure

Please add information how the microcosms were managed in term of temperatures, soil humidity and light intensity

Do you have a control with microcosms with the application of synthetic fertilizers?

Add statistical differences in tables 1 and 2 for the investigated amendment

Add the standard error in the tables and figures

Author Response

Response to Reviewer #2 comments:

This is an interesting study, and it is easy to read.

Thank you very much for your positive comments.

Here some suggestions:

In the introduction put the attention not only on Argentina but extend the area at world level, at least at aera with similar condition to that one investigated in the present research, on the other hand put the word “Argentina” in the title.

Thank you for the  comment. We agree. We have added the word “Argentina” in the title as per your suggestion.

Add the kg of feeding using per cow

Thank you for the  comment. We agree. We have added the detailed information in the section 2.1 Diet per cow per day is composed of 45 % of pasture (9 kg), 22% of maize silage (4 kg) and 33% of concentrate (6 kg; maize and soybean meal), averaged over the year.

Explain better the use of onion by-products and why in combination with the manure

Thank you for the  comment. We agree.  We have added more detailed information addressing these questions in section 2.2.

Please add information how the microcosms were managed in term of temperatures, soil humidity and light intensity

Thank you for the  comment. We agree.  We have included additional details in section 2.3.1

Do you have a control with microcosms with the application of synthetic fertilizers?

Thank you for the comment. We did not include synthetic fertilizers in this experiment.

Add statistical differences in tables 1 and 2 for the investigated amendment

Thank you for the  comment. We agree. The characterization of these amendments is considered baseline data due to their different origin and processing; we did not pursue any descriptive statistical measures besides mean values. 

Add the standard error in the tables and figures

Thank you for the comment. We have added all the standard errors in the supplementary tables. For figures, we prefer to keep using the LSD values (as bar plot) to facilitate the visual comparison of treatments at different times. The interpretation of these LSD bars for mean comparison is now explained in the statistical analysis section (2.4).